# Social and environmental impact of recent developments in machine learning on biology and chemistry research

## Abstract

Potential societal and environmental effects such as the rapidly increasing resource use and the associated environmental impact, reproducibility issues, and exclusivity, the privatization of ML research leading to a public research brain-drain, a narrowing of the research effort caused by a focus on deep learning, and the introduction of biases through a lack of sociodemographic diversity in data and personnel caused by recent developments in machine learning are a current topic of discussion and scientific publications. However, these discussions and publications focus mainly on computer science-adjacent fields, including computer vision and natural language processing or basic ML research. Using bibliometric analysis of the complete and full-text analysis of the open-access literature, we show that the same observations can be made for applied machine learning in chemistry and biology. These developments can potentially affect basic and applied research, such as drug discovery and development, beyond the known issue of biased data sets.

## 1 Introduction

The unprecedented progress of machine learning during the past two decades has been catalysed and remains driven by the development of increasingly powerful computer hardware. This progress is enabled by the ability of deep neural networks to scale exceptionally well with increasing data availability and model complexity compared to other approaches. Thus, they can be trained for linear regression on small data sets and, with conceptually simple changes to the network architecture, for language translation or image generation on immense text corpora and image collections. While comparatively exceptional, deep neural networks are understood to still only scale linearly at an exponential cost (Schwartz et al., 2020), leading to diminishing returns (Thompson et al., 2021). Among the machine learning community, this has raised concern over the future direction of the field and a growing exclusivity driven by ever-increasing hardware and energy costs (Schwartz et al., 2020; Thompson et al., 2021; Jurowetzki et al., 2021). After a discourse on the intertwined recent history of deep learning and hardware advances, we will analyse the applicability of the most prominent concerns raised in machine learning research to applied machine learning research in biology and chemistry. We have categorised these concerns under socioeconimic, scientific, and environmental considerations.

**The hard- and software that catalysed rapid developments in machine learning**    In late 2002 and early 2003, the release of the Radeon 9700 and GeForce FX video cards introduced a fully programmable graphics pipeline, extending and later replacing the existing fixed function pipelines. Unlike the fixed function pipeline, which allowed the user to only supply input matrices and parameters to built-in operations, the programmable pipeline introduced the execution of user-written shader programs on the GPU (Contributors, 2015). This fundamental change allowed programmers and researchers to exploit the intrinsic parallelism of GPUs 2 years before Intel would introduce its first dual-core CPU. Within months of the availability of this new hardware and the accompanying APIs, researchers implemented linear algebra methods on GPUs and introduced programming frameworks to use GPUs for general-purpose computations (Thompson et al., 2002; Krüger & West-

ermann, 2003). This rapid development marked the dawn of general-purpose computing on graphics processing units (GPGPU). In a presentation at ICS '08, Harris presented the successes of GPGPU by highlighting a speed-up in molecular docking, N-body simulations, HD video stream transcoding, or image processing—applications in machine learning were not discussed. However, just one year later, the introduction of GPUs as general-purpose processors catalysed the deep learning explosion of the early 2010s by allowing deep learning algorithms pioneered by Alexey Ivakhnenko in 1971 to be run within practical time on widely available consumer hardware when Rajat et al. showed that GPUs outperform CPUs by an order of magnitude in large-scale deep unsupervised learning tasks (Ivakhnenko, 1971; Raina et al., 2009).

**Hardware and energy requirements increase in machine learning research**    In 2010, Ciresan et al. (2010) introduced a multi-layer perceptron (MLP) with up to 12.11 million free parameters where forward and backward propagation were implemented on a GPU using NVIDIA's proprietary CUDA API introduced by Harris at ICS '08 two years before, speeding up the routines by a factor of 40. In their arXiv paper, they also report the computer's hardware specifications as "Core2 Quad 9450 2.66GHz processor, 3GB of RAM, and a GTX280 graphics card". The GTX 280 graphics card by NVIDIA was, at the time of the paper's writing, two years old and cost USD 893 when first released (adjusted for inflation). Equipped with this 2-year-old hardware that cost well below USD 1,000, Cireşan et al. were able to improve upon the state-of-the-art performance on the MNIST classification benchmark set four years prior by Ranzato et al. (2006). As they not only reported the hardware used but also the time it took to train the model, the power usage of the GPU and CPU, with a thermal design power (TDP) of 236 and 95 Watt, respectively, can be calculated as 114.5h $\times$ (236 + 95)W = 37.9kWh. Seven years later, Vaswani et al. (2017) introduced the transformer architecture. The training used 8 NVIDIA Tesla P100 GPUs, whose price was $\sim$USD 55,100 at the time, and took 84 hours, resulting in overall energy usage of 84h $\times$ 8 $\times$ 250W = 168kWh.

**Hardware and energy requirements explode in applied machine learning**    Applying the novel transformer architecture, NVIDIA reportedly trained the 345 million parameter BERT model in 2019, which was previously introduced by Google in the same year, on 4 DGX-2H servers (64 Tesla V100s) in 79.2 hours, with a maximum power usage of 12,000 Watt, resulting in a total power use of 3.8 MWh (79.2h $\times$ 4 $\times$ 12kW) (Devlin et al., 2018). The cost of this system at the time of training was USD 1,596,000. Alternatively, the BERT model could be trained on on-demand Google Cloud GPUs for USD 2.48 per GPU hour, resulting in total costs of USD 12,570 (2.48 $\times$ 64 $\times$ 79.2). The MT-NLG model presented by NVIDIA and Microsoft in 2021 represents the acceleration of hardware and energy cost in the field (Smith et al., 2022). The 530 billion parameter model was trained on 560 DGX A100 servers—a total of 4,480 NVIDIA A100 80GB Tensor Core GPUs—for 2,160 hours (Rajbhandari et al., 2022). The power usage of a cluster of 560 DGX A100 servers during 2,160 hours is 7.862 GWh (2,160 $\times$ 560 $\times$ 6.5kW). Taking the world average electricity price of USD 0.131 per kWh during December 2021, the total electricity bill for training MT-NLG was USD 1,029,922. The total cost of the hardware is hard to estimate as specialised network hardware is required to build such a cluster; however, the DGX A100 was priced at USD 199,000 on release, resulting in a minimum total cost of USD 111,440,000 (199,000 $\times$ 560). Training the model on on-demand Google Cloud GPUs for USD 2,141.82 per GPU month results in a total cost of USD 28,786,060.8 (3 $\times$ 4,480 $\times$ 2,141.82).

**Hardware and energy costs drive the de-democratization of machine learning**    The examples discussed above represent an increasing hardware and energy cost in conducting basic and applied deep learning-based machine learning research. The resulting diminishing returns and the environmental impact have previously been discussed by Thompson et al. (2021) and Schwartz et al. (2020). The development of increasing costs following a potential breakthrough stands in contrast to similar or even more disruptive changes in other fields, such as CRISPR-Cas9 lowering costs in molecular biology or the ever-decreasing costs of genome and RNA sequencing (Ledford, 2015; Wetterstrand, 2021; Gierahn et al., 2017). While CRISPR-Cas9 and affordable sequencing has led to what has been called the democratization of access to sequencing and genome editing (Guernet & Grumolato, 2017; McPherson, 2014; Srivathsan et al., 2019), cutting-edge machine learning research is becoming potentially increasingly expensive and exclusive (Ahmed & Wahed, 2020). Indeed, in a 2020 article on the most cited research articles, all mentioned machine learning research was conducted by, or in collaboration with, OpenAI, Microsoft, and Alphabet (Kingma & Ba, 2014; Ren et al.,

2015; Mnih et al., 2015; Vaswani et al., 2017; Silver et al., 2016; Crew, 2020), suggesting a need for resources not available within academia. While the involvement of these companies, whose R&D budgets exceed those of most nations (Bajpai, 2021; Ballard, 2021; Uis, 2022), contributed greatly to the advancement of machine learning, a potential technological dependency of academia on this commercially driven progress combined with a brain-train from academia to industry may result in a narrowing of machine learning research (Jurowetzki et al., 2021; Klinger et al., 2020; Hagendorff & Meding, 2020). In the following paragraph we will discuss these developments and the concerns raised with a focus on applied machine learning in chemistry and biology.

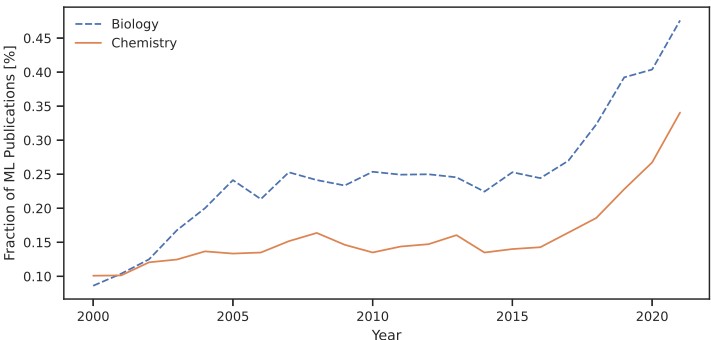

Figure 1: Fraction of applied machine learning publications in biology and chemistry. After an increase in the early 2000s and a plateau lasting for 10 years, the introduction of general-purpose GPUs and a renewed interest in deep neural networks making use of this capable hardware sparked a unprecedented growth of machine learning research in biology and chemistry in the mid-2010s.

## 2 SOCIOECONOMIC CONSIDERATIONS

Available research funding differs greatly between countries and institutions. While discussing the reasons and possible solutions to this situation are far beyond the scope of this research, it is important to note that cost of research being driven by ever increasing hardware and electricity requirements has the potential to further drive inequity between nations, institutions, research groups, and individual researchers based on available funding (Gerhards et al., 2018; Nielsen & Andersen, 2021; Wahls, 2018). In addition to direct funding, collaborations with leading big tech companies, which led to the most cited recent publications (Crew, 2020), are equally unevenly distributed among countries and institutions, further concentrating funding. Furthermore, the availability of necessary hardware to conduct state-of-the-art deep learning research can depend on the current geopolitical situation, possibly disproportionately affecting researchers in low- and middle-income countries negatively (Nellis & Lee, 2022). In this section, we discuss social and economic concerns raised within the machine learning community, including machine learning in a divided society and that diminishing returns limit participation.

**Machine learning in a divided society**   In a wider social context, the concentration of resources and talent needed to develop and operate machine learning systems in high-income countries form the basis of what has been labelled colonialist AI (Birhane, 2020; Mohamed et al., 2020; Adams, 2021). The effects of this process recently came to light through the discovery of biases in models in healthcare and policing, among others (Obermeyer et al., 2019; Hildebrandt et al., 2020). Causes of these biases have been shown to go beyond biased data (Schwartz et al., 2022; Mehrabi et al., 2021).

These effects introduced above are of interest for the natural sciences, concretely biology and chemistry, as deep learning-based applied machine learning plays an ever more critical role in both fields and issues being caused by said effects may affect crucial research such as drug development or genetics. The increasing importance of machine learning in biology and chemistry is reflected by an increasing share of machine learning-related publications, which have approximately doubled within the past six years compared to the overall published literature in the respective fields (Figure 1). Here, the potential for the described societal and environmental effects within the scientific

fields of biology and chemistry will be explored based on quantitative bibliometric analysis. The utilised data was downloaded from the OpenAlex index of scholarly works (Priem et al., 2022). The final data set contained 27,861 biology and 16,301 chemistry works filtered for machine learning topics. For a list of filter keywords, see Appendix A.

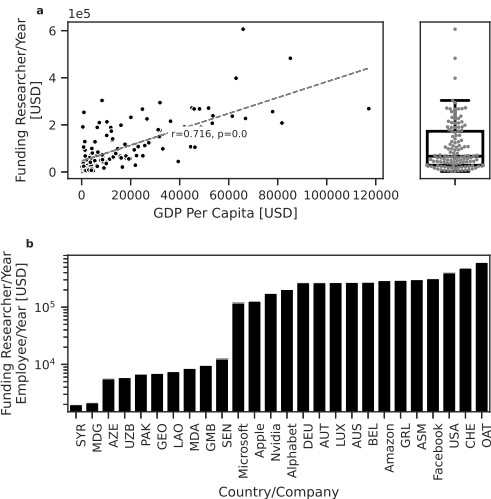

Figure 2: R&D funding of nation states and "big tech" companies. (a) R&D funding per researcher and year correlates well with a countries GDP. (b) The amount of R&D funding per researcher and year differs greatly between countries. Comparing the R&D budgets per employee (includes non-research employees) of "big tech" companies places them among the top funded nations.

**Diminishing returns that limit participation**    As current and future diminishing returns of deep learning in basic and applied machine learning research threaten to drive costs up, the financial barrier to participating in the respective fields also increases (Thompson et al., 2021; Schwartz et al., 2020). Given the extreme disparities in R&D funding across countries (Figure 2) and institutions has great potential to also increase the exclusiveness of participating in state-of-the-art research. Indeed, potential downstream effects of such disparities have recently come in the focus in fields such as health care, bioinformatics, and biometrics, where machine learning models are often biased towards persons of European descent (Obermeyer et al., 2019; Gupta et al., 2020; Celi et al., 2022; Dias & Torkamani, 2019; Kiseleva & Quinn, 2021; Drozdowski et al., 2020; Hazirbas et al., 2022). While these issues are being tackled by increasing participation of students, developers, and researchers of non-European descent and by debiasing data sets, it is being questioned whether these efforts alone can adequately solve the bias-problem and avert a future long-term dependence of the global south on the global north (Birhane, 2020; Bon et al., 2021).

In respect to machine learning in biology and chemistry, we make the assumption that an increase in the correlation between available funding and the share of machine learning literature published in each field would suggest a potential for exclusiveness and a measure for inequity. In order to identify a possible increase in correlation between available R&D funding and applied machine learning in biology and chemistry, as well as the impact of deep learning-based machine learning, 137,506 affiliations from bioinformatics/computational biology and 80,206 affiliations from cheminformatics/computational chemistry publications were analysed for the periods between 2000 to 2016 and 2017 to 2021 (Figure 3). Only countries with at least 100 publications per year during the specified periods were included, and outliers were removed using a z-score cut-off of $\pm 3$. Plotting the fraction of machine learning publications amongst the entire published literature for the chosen topics from biology and chemistry for each country against the countries' available funding per researcher shows an increasing, however as of yet insignificant, correlation. While the average fraction of machine learning literature increased by 19.3% (21.7% to 41%) and 9% (20.4% to 29.4%) in biology and chemistry, respectively, the Gini coefficient of the fraction of machine learning publications remained stable in biology (0.287 to 0.287) and decreased in chemistry (0.343 to 0.304). The fund-

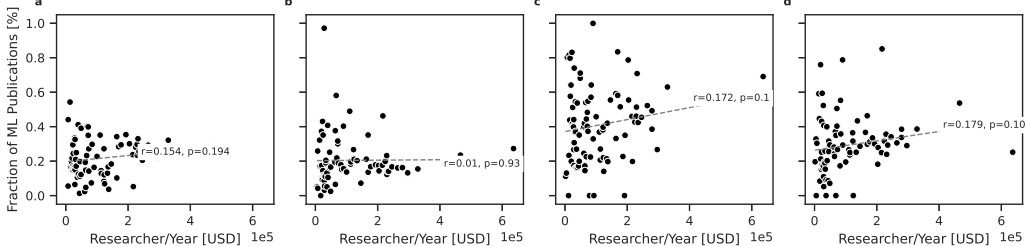

Figure 3: Fraction of applied machine learning papers in biology and chemistry as a function of R&D funding per researcher and year. (a) Biology 2000–2016, (b) chemistry 2000–2016, (c) biology 2017–2021, (d) chemistry 2017–2021. While not significantly so, a potential trend is emerging where the fraction of applied machine learning publications (of each fields total) correlates with the funding per researcher and year.

ing per researcher and year is an average of the available data between 2000 and 2021 for each country—with an average Gini coefficient of 0.479 (0.452–0.497) over all stratifications.

## 3 SCIENTIFIC CONSIDERATIONS

Different concerns with a potential effect on the scientific community have been raised within the machine learning field. In this section, we analyse the most prominent of these concerns including the privatization of machine learning, a possible academic brain-drain, citation inequality, and a narrowing of research.

**The privatization of machine learning** Beyond Universities, technology companies are major contributors to basic and applied deep learning-based machine learning research across multiple fields. Prominent examples are AlphaFold by DeepMind, BERT by Google, 3D human pose estimation by Facebook and Google, or DALL·E by OpenAI (Jumper et al., 2021; Devlin et al., 2018; Ramesh et al., 2021). In addition, the hardware company NVIDIA, which GPUs and CUDA framework power most basic and applied deep learning research (with the notable exception of research conducted by Google and DeepMind, both using Google's TPUs), is active in both basic research and deep learning applied to computer graphics (Karras et al., 2020; Wang et al., 2019). The involvement of these corporations in research is significant, as their respective R&D budgets are higher than those of most countries (Figure 2b). For 2020 (2021 for NVIDIA), the R&D spending of Google (now Alphabet) was USD 27.57B, that of Facebook USD 18.45B, and that of NVIDIA USD 3.92B and were higher than that of 120, 116, and 101 countries, respectively (Bajpai, 2021; Ballard, 2021). For Google and Facebook, this includes countries such as Belgium, Spain, India, or the Russian Federation (Uis, 2022). The availability of this large amounts of funding for machine learning research in industry can have three significant effects on public university research:

(i) During the past two decades, there has been an unprecedented brain drain of professors from university to industry, a development that has been termed the "privatization of AI research(-ers)" (Gofman & Jin, 2019; Jurowetzki et al., 2021).

(ii) Industry collaborations that provide access to significant amounts of compute are often limited to elite institutions in countries where funding is already comparatively high, further increasing the divide in resource availability between institutions within a country and globally (Jurowetzki et al., 2021; Ahmed & Wahed, 2020).

(iii) A narrowing of research driven by universities aligning with commercial interest through collaborations, a reliance of universities on models, methods, and hardware developed in industry, and the above mentioned brain-drain. (Klinger et al., 2020; Gofman & Jin, 2019).

Potential consequences of these developments have been shown to include stagnation of (methodical) diversity in machine learning research with a preference for compute-intensive and data-hungry deep learning methods (Klinger et al., 2020). In addition, an analysis of research pre-

sented at NeurIPS, CVPR, and ICML showed that researchers in the field often fail to report conflicts of interest, that publications from the industry gather significantly higher citations than those from academia, and that industry publishes on trending machine learning topics two years before academia on average (Hagendorff & Meding, 2020).

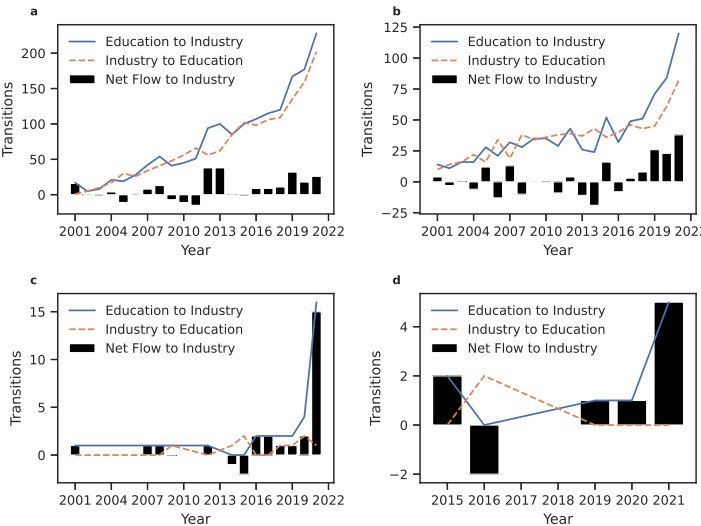

Figure 4: Transitions by researchers between education and industry. (a) In biology, the overall transitions between academia and industry remain largely balanced over the observed period. (b) In chemistry, we observe a recent steady increase of transitions from academia to industry. (c) Starting in 2016, there is a steady net flow from acadamia to *Tech*, spiking in 2021. (d) A similar but less pronounced spike can be observed in chemistry.

In order to assess the existence of such effects in biology and chemistry, the available data was labelled by affiliation-type. Industry affiliations were labelled with either *Tech*, *Pharma*, *Chem*, and *Other*. *Tech* includes the large technology companies (Alphabet, Microsoft, Meta, Nvidia, Amazon, IBM, Tencent, Baidu, and Twitter) and their subsidiaries. *Pharma* includes pharmaceutical companies such as Novartis, Pfizer, and Bayer. *Chem* includes chemical and petrochemical companies such as BASF, Exxon, and Lonza. *Other* includes companies not falling into the other categories, such as Schrödinger, General Electrics, or Siemens. All other affiliations were labelled with *None*.

**Academic brain-drain** Following the approach by Jurowetzki et al. (2021), transitions between academia and industry were defined as a year-over-year change in affiliation (if there were industry as well as academic affiliations within the same year, the mode was chosen as the overall affiliation). In the results, based on 92,496 and 54,243 authors of biology and chemistry articles, respectively, trends that are similar to those observed by Jurowetzki et al. (2021) can be noticed (Figure 4). Over the past six years, chemistry saw an unprecedented increase in net transitions of machine learning practitioners from academia to industry (Figure 4b), while overall transitions in biology remain slightly skewed towards industry (Figure 4a). However, biology, and to lesser degree chemistry, saw a significant jump in academia to *Tech* transitions in 2021 (Figure 4c,d). Interestingly, the data shows a lower baseline of academia to industry transitions compared to the data presented by Jurowetzki et al. (2021), suggesting either fewer transitions from academia to industry overall or, more likely, a tendency in the fields to stop publishing after leaving academia.

**Citation inequality** The observation by Hagendorff & Meding (2020) of significantly higher rates of citations of papers authored by industry-affiliated authors can be replicated for machine learning in chemistry and partially for machine learning in biology. In biology, citation rates by *Pharma*, *Tech*, and *Other* are significantly higher than those in academia (with a mean of 6.31, 21.64, and 4.13 versus 3.03 citations per year, respectively. Figure 5a). In chemistry, the citations per year are significantly higher for *Pharma*, *Chem*, *Tech*, and *Other*, compared to academia (with a mean of 4.46, 4.00, 8.71, and 3.62 versus 2.99, respectively. Figure 5g). Breaking down the affiliations

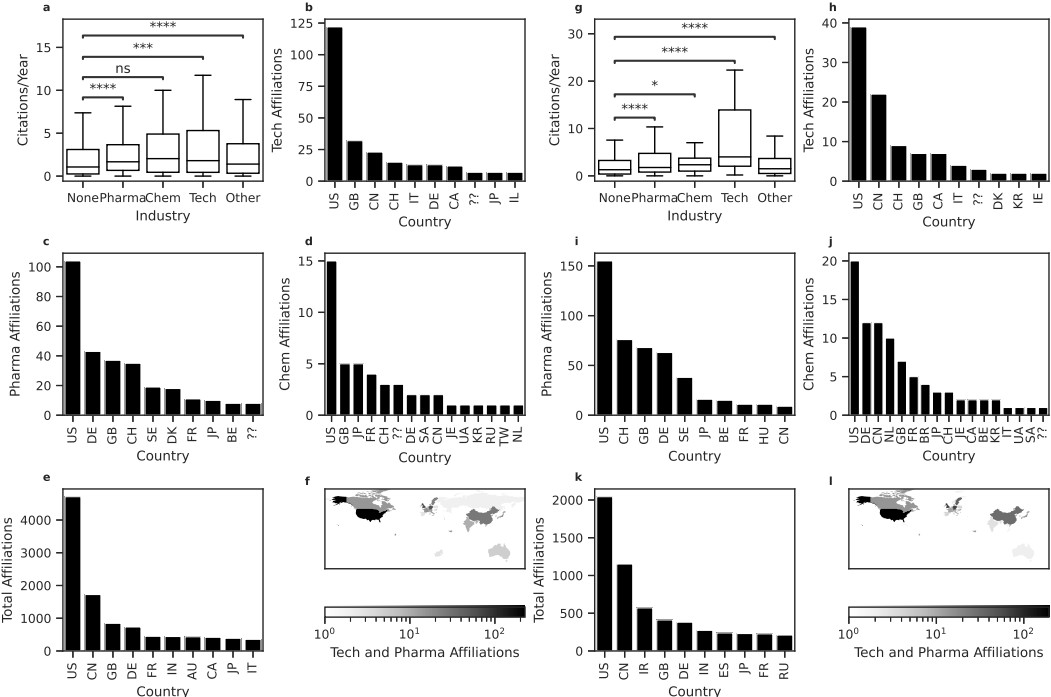

Figure 5: Industry affiliations and the effect on citations. For both biology (a) and chemistry (g), publications with industry affiliations gather a higher amount of citations per year. This effect is, in both fields, most pronounced for *Tech* affiliations. Breaking down the industry affiliations by country (Figure 5b–f for biology, h–l for chemistry), it is little surprising that private research generally takes place in countries with high ranking universities and highly developed industries.

by country, it is little surprising that private research generally takes place in countries with high-ranking universities and highly developed industries (Figure 5b–f for biology, h–l for chemistry).

**A narrowing of research** A narrowing of machine learning research would certainly have the potential to lead to negative downstream effects in the natural sciences. In addition, there is also the potential for a narrowing of applied methods independently from developments in machine learning research. Such a narrowing of applied methods can be driven by the same factors as a narrowing in basic machine learning research. These factors include demand-side economies of scale or a bandwagon effect, early fortuitous events that result in lock-in to a certain method, the availability of and previous investments in suitable resources such as hardware or personnel, and social dilemmas (Klinger et al., 2020). To explore a potential narrowing of machine learning methods utilised in biology and chemistry, articles have been labelled with the following broad categories: *Dimensionality Reduction/Feature Selection*, *Genetic/Evolutionary*, *Neural Networks*, *Other/Unspecified*, *Regression*, *Statistic/Probabilistic*, and *SVM*. Analysing the temporal development, the results show that the largely deep learning-based *Neural Networks* in biology (Figure 6a) and *Neural Networks* together with *RF/Boosting* in chemistry (Figure 6b) are mainly responsible for the recent explosive growth of machine learning literature in the two fields. However, the continuous and roughly linear growth of the other categories implies that the fast-growing methods do not seem to grow at the expense of methods from other categories. Plotting the mean of the citations per year for each category shows that the publication of highly cited biology articles utilising *Neural Networks* predates and is more sustained (Figure 6c) than the publication of highly cited chemistry articles utilising methods from the same category (Figure 6d).

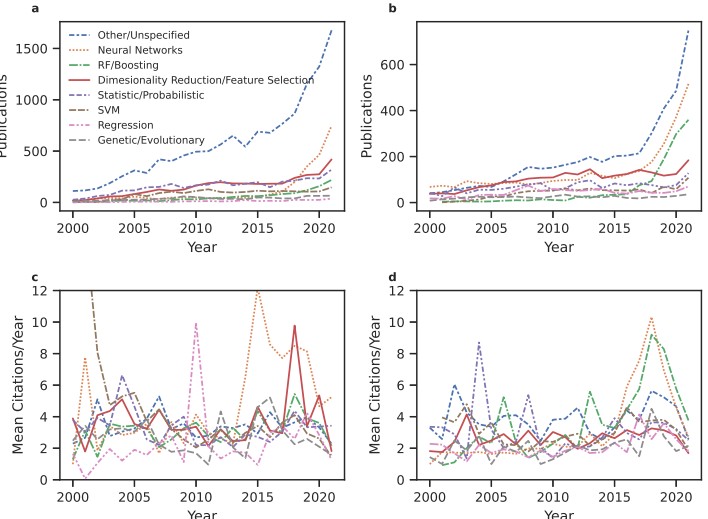

Figure 6: Number of publications by machine learning method category. (a) Number of publications by machine learning method category in biology. (b) Number of publications by machine learning category in chemistry. (c) Mean citations/year by year and machine learning category in biology. (d) Mean citations/year by year and machine learning category in chemistry.

## 4 ENVIRONMENTAL CONSIDERATIONS

In their publication "Green AI", Schwartz et al. (2020) showed that the carbon footprint of deep learning has been growing continuously and suggest making efficiency an evaluation criterion in addition to accuracy. They argue that this ever-increasing need for more compute would make deep learning-based machine learning research not only environmentally problematic but also less inclusive due to the previously discussed unequal distribution of access to compute.

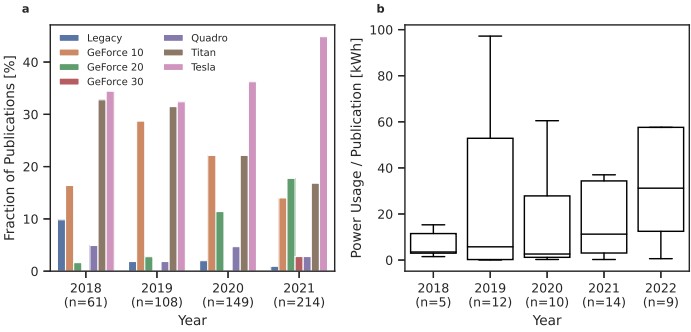

Figure 7: Hardware and energy use of applied machine learning in biology and chemistry. (a) GPU models used by biology and chemistry researchers combined, showing an increase in the fraction of machine learning-dedicated Tesla series GPUs used, while the share of consumer and enthusiast oriented GPU series is shrinking. (b) The data on training times and exact GPU model that could be extracted from open access literature shows a general increase in electricity use per publication.

The quantitative exploration of the environmental impact of applied machine learning in biology and chemistry has been complicated by the low availability of information regarding the hardware used and the time spent to train the models. From a total of 27,861 biology and 16,301 chemistry articles, 14,365 (51.6%) and 4,184 (25.7%) were downloadable as open access articles. From these downloaded articles, the GPU model could be extracted from 646 (4.5%) biology articles and from

243 (5.8%) chemistry articles, and the GPU model as well as the training time from 56 (0.4%) and 14 (0.3%), respectively. Based on this sparse data, the statistical power of the analysis is relatively low. However, certain trends in the GPU model-use can be observed between 2018 and 2021 (Figure 7a): The fraction of Nvidia Titan GPUs has been steadily declining during the entire period. Consumer-focused GPUs from the GeForce 10, 20, and 30 series continue to be used in research, with the fraction of GeForce 20 GPUs increasing. In contrast, GeForce 10 series cards experienced a decline starting in 2019. However, the most important trend is the increase in the fraction of Tesla GPUs.

As newer GPU models are becoming more energy efficient (Špeťko et al., 2021), efficiency remains a second-class metric compared to accuracy in applied machine learning publications in biology and chemistry. This situation is reflected in the lack of an increase in reported hardware use or training times (Figure 7b). Indeed, the available data shows a general increase in power usage per publication based on the reported GPU model, the number of GPUs used, and the training time. The values are an estimation and only represent the training time of the final model.

## 5 CONCLUSION

The societal and environmental impacts of recent developments in machine learning are actively being discussed in machine learning research and closely adjoined fields such as natural language processing or image processing. However, the effects driven by and experienced within applied machine learning in the natural sciences received little attention. This exploratory study into the social and environmental impact of recent developments in machine learning on biology and chemistry research brings to light the following insights:

(i) The introduction of deep learning methods has caused a rapid increase in the share of machine learning-related articles in biology and chemistry literature.

(ii) There is a potential emerging trend towards an increase in inequity in applied machine learning research conducted in biology and chemistry based on available funding when corrected for overall publications in the respective field. While not yet statistically significant, the trend should be monitored and countered. There is, however, a significant difference between citation metrics of education vs industry affiliated researchers, with university-affiliated research receiving fewer citations.

(iii) Based on transition patterns of researchers between academia and industry, big tech companies seem to get increasingly involved in conducting applied machine learning research in biology and chemistry. Given the vast resources and R&D spending of these corporations (even compared to nations), the concerns of a potential brain-drain away from academia and a narrowing of the research conducted are also warranted in biology and chemistry.

(iv) As of yet and based on the number of publications, the growth of neural networks-based deep learning research did not happen at the expanse of other, generally less compute-intensive, categories of machine learning methods. However, citation metrics for neural network-based deep learning methods have recently spiked compared to other categories. While this data does not show a narrowing of the use of methods, continuous monitoring of these metrics to avoid a potential method and technology lock-in, especially as investments in specialized hardware continue, may be warranted.

(v) Data that allows an estimation of the environmental impact of applied machine learning in biology and chemistry is scarce due to a failure to report used hardware and spent training time in the vast majority of articles. This scarcity shows the need for an effort by both authors and publishers to introduce measures concerning reporting these metrics, especially as their importance goes beyond monitoring the environmental impact. Nevertheless, based on the available data, a trend towards more specialized hardware and an overall increase in energy use per paper could be observed.

Overall, these insights are similar to the ones made in machine learning research in general and warrant action, given that research topics with a high societal impact, such as genetics or drug development, can be affected. While each topic may be discussed in detail in future publications, this overview provides the basis for discussion and draws attention to the crucial interfaces between society, the economy, the natural sciences, and machine learning.

## 6 Reproducibility Statement

All data used in this study is freely accessible. The code to reproduce that data as well as the figures can be found in the following anonymised repository: `https://anonymous.4open.science/r/anon_aichem-83F8`

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

# A APPENDIX

## A.1 OPENALEX QUERIES

The OpenAlex database was queried for works of type *journal-article* and *proceedings-article* and filtered on machine learning concepts and biology or chemistry concepts.

The query for machine learning works in biology was `https://api.openalex.org/works?per-page=200&cursor=*&filter=concepts.id:C154945302|C119857082|C108583219|C50644808|C8880873|C159149176|C126980161|C58328972|C97541855|C12267149|C179717631|C81363708|C147168706|C46686674|C169258074,concepts.id:74187038|C64903051|C32909587|C177801218|C99726746|C63222358|C103697762|C192071366|C164126121|C69366308|C185592680|C147597530|C59593255|C161790260|C178790620|C98274493,type:journal-article|proceedings-article.`

The query for machine learning works in chemistry was `https://api.openalex.org/works?per-page=200&cursor=*&filter=concepts.id:C154945302|C119857082|C108583219|C50644808|C8880873|C159149176|C126980161|C58328972|C97541855|C12267149|C179717631|C81363708|C147168706|C46686674|C169258074,concepts.id:C86803240|C60644358|C54355233|C70721500|C140556311|C153911025|C78458016|C95444343|C89423630|C203014093|C21565614|C46111723|C157585117|C191120209|C55493867|C204328495,type:journal-article|proceedings-article.`

The query for all biology works was: `https://api.openalex.org/works?per-page=200&cursor=*&filter=concepts.id:C74187038|C64903051|C32909587|C177801218|C99726746|C63222358|C103697762|C192071366|C164126121|C69366308|C185592680|C147597530|C59593255|C161790260|C178790620|C98274493,publication_year:<year>,type:journal-article|proceedings-article&group_by=authorships.institutions.country_code`

The query for all chemistry works was: `https://api.openalex.org/works?per-page=200&cursor=*&filter=concepts.id:C86803240|C60644358|C54355233|C70721500|C140556311|C153911025|C78458016|C95444343|C89423630|C203014093|C21565614|C46111723|C157585117|C191120209|C55493867|C204328495,publication_year:<year>,type:journal-article|proceedings-article&group_by=authorships.institutions.country_code`

- Machine learning concepts
    - Artificial Intelligence: C154945302
    - Machine Learning: C119857082
    - Deep Learning: C108583219
    - Artificial Neural Network: C50644808
    - Genetic Algorithm: C8880873
    - Evolutionary Algorithm: C159149176
    - Simulated Annealing: C126980161
    - Expert System: C58328972
    - Reinforcement Learning: C97541855
    - Support Vector Machine: C12267149
    - Multilayer Perceptron: C179717631
    - Convolutional Neural Network: C81363708
    - Recurrent Neural Network: C147168706
    - Boosting: C46686674
    - Random Forest: C169258074
- Biology concepts
    - Biology: C86803240

- – Bioinformatics: C60644358
- – Genetics: C54355233
- – Computational Biology: C70721500
- – Biostatistics: C140556311
- – Molecular Biology: C153911025
- – Evolutionary Biology: C78458016
- – Cell Biology: C95444343
- – Microbiology: C89423630
- – Immunology: C203014093
- – Metabolomics: C21565614
- – Proteomics: C46111723
- – Omics: C157585117
- – Structural Biology: C191120209
- – Biochemistry: C55493867
- – Protein Folding: C204328495
- Chemistry concepts
  - – Drug Discovery: C74187038
  - – Drug Design: C64903051
  - – Molecule: C32909587
  - – Chemical Reaction: C177801218
  - – Chemical Space: C99726746
  - – chEMBL: C63222358
  - – Virtual screening: C103697762
  - – Drug Design: C192071366
  - – QSAR: C164126121
  - – ADME: C69366308
  - – Chemistry: C185592680
  - – Computational Chemistry: C147597530
  - – Molecular Dynamics: C59593255
  - – Catalysis: C161790260
  - – Organic Chemistry: C178790620
  - – Pharmacology: C98274493

