# OpenReview forum: "Social and environmental impact of recent developments in machine learning on biology and chemistry research"
_ICLR.cc/2023/Conference — Submitted to ICLR 2023_

### Official Review · Reviewer_s4UR · 2022-10-18

**Confidence:** 4
**Correctness:** 3
**Technical Novelty And Significance:** 2
**Empirical Novelty And Significance:** 2
**Recommendation:** 5

**Clarity, Quality, Novelty And Reproducibility:**


All experiments seem to be reproducible.

**Strength And Weaknesses:**


The topic is of importance and in general interesting.

**Summary Of The Paper:**


In this paper, the authors perform several quantitive studies for understanding the social and environmental impact when applying machine learning methods for biology and chemistry research. Some of authors observations include a potential inequity increase in applied machine learning research in biology and chemistry, and big companies tend to be actively involved in machine learning based research.



**Summary Of The Review:**


This paper is not proposing some new tasks or introducing some new methods. Although the topic and the authors' findings are interesting, it seems this paper is not suitable for a conference like ICLR.

---

> ### Author Response · Authors · 2022-11-12
> **Regarding the suitability for a conference like ICLR**
>
> Dear reviewer,
>
> Thank you for your input.
>
> ### Summary Of The Review
> Regarding the suitability for a conference like ICLR: We decided to submit this paper, which we initially would have agreed to not fit into what we had commonly read at ICLR, because we found the list of *relevant topics explored at the conference* on the [conference page](https://iclr.cc/). This list includes the item **societal considerations of representation learning including fairness, safety, privacy, and interpretability, and explainability**. Based on this, we made the decision to submit the manuscript to ICLR, as I think it makes a relevant contribution to this topic. Let us know if you understand this differently.
>
> We hope that you would reconsider your recommendation based on this feedback. You can also find an updated version of the manuscript that includes restructurings and additional introductory paragraphs based on the feedback on other reviewers.
>
> The authors.

---

### Official Review · Reviewer_RDG6 · 2022-10-21

**Confidence:** 3
**Correctness:** 3
**Technical Novelty And Significance:** 2
**Empirical Novelty And Significance:** 3
**Recommendation:** 3

**Clarity, Quality, Novelty And Reproducibility:**

Not very clear for the reasons stated above.
Definitely novel from the point of vew of the research and certainly reproducible.

**Strength And Weaknesses:**

Strengths:
Lots of information and very interesting topic
Weaknesses:
Minimally coherent structure. At some points it almost seems like a stream of thoughts dump.
Mixed topics
Unorthodox information source referencing

**Summary Of The Paper:**

Paper studying social and environmental impact of recent developments in ML (with a focus on DL) on biology and chemistry research.

**Summary Of The Review:**

Interesting paper studying social and environmental impact of recent developments in ML (with a focus on DL) on biology and chemistry research.

It falls within the remit of the conference, as the call for papers includes work on "societal considerations of representation learning
including fairness, safety, privacy, and interpretability, and explainability"

Even if interesting and certainly relevant, my main qualm with this study is that it lacks coherent structure. A lot of ideas and information
are thrown in the pot, but they are barely organized at all, jumping from a topic to the next, almost without the respite of a paragraph ending.
This is compounded by a rather unirthodox referencing system.

I definitely like the content, but it seems to me that it would require a different venue in which these ideas could breath, be it a jounal position paper or a book chapter.

---

> ### Author Response · Authors · 2022-11-12
> **Updated the structure and added introductory paragraphs.**
>
> Dear reviewer,
>
> Thank you for your feedback.
>
> ### Strength And Weaknesses
> #### Weaknesses
> Regarding the *Unorthodox information source referencing*, may we ask what is meant by that? If it concerns the references to news articles, this is on one hand due to a lack of financial reporting in the academic literature and on the other hand, a lack of transparency regarding the reporting of hardware and time used to train large models.
>
> The overall organisation of the paper is an analysis of different prominent concerns raised within the machine learning research community applied to machine learning in biology and chemistry. The manuscript has been updated to make this clearer by adding additional introductory paragraphs to the introduction and the three main sections. Furthermore, we have reorganised the paragraphs to increase the overall readability.
>
> We hope that this answers your major concerns. You can find all the changes in the updated pdf document.
>
> The authors.

---

### Official Review · Reviewer_6kvg · 2022-10-24

**Confidence:** 4
**Correctness:** 4
**Technical Novelty And Significance:** 3
**Empirical Novelty And Significance:** 3
**Recommendation:** 8

**Clarity, Quality, Novelty And Reproducibility:**

The paper presents novel knowledge as it specifically draws attention to the impact of machine learning research in biology and chemistry. The findings are also general in  that they apply to other fields as well.
The results reported in the study may be reproduced as the authors have shared  data  and code used in this study.

**Details Of Ethics Concerns:**

None.

**Strength And Weaknesses:**

Strength: results reported were consistently compared with those in literature.

Weakness 1: The sentences are just too long. Need to be broken down. For example, this sentence is too long : “Potential societal and environmental effects such as the rapidly increasing resource use and the associated environmental impact, reproducibility issues, and exclusivity, the privatization of ML research leading to a public research brain-drain, a narrowing of the research effort caused by a focus on deep learning, and the introduction of biases through a lack of sociodemographic diversity in data and personnel caused by recent developments in machine learning are a current topic of discussion and scientific publications.

Weakness 2: A simple figure highlighting all the steps/methods that were followed in generating the paper findings is missing.



**Summary Of The Paper:**

The paper draws attention to the societal and environmental impacts of machine learning research in chemistry and biology. The paper specifically makes a contribution in the following components such as: environmental considerations, citation inequality, academic brain-drain, and scientific considerations:

The study makes the following key assertions:

* During the past two decades, there has been an  unparalleled shift of professors from university to industry.
* chemistry saw an increase in transitions of machine learning practitioners from academia to industry, while overall transitions in biology remain slightly skewed towards industry
* Technology companies drive machine learning research.
* More citations in chemistry and biology are generated from Pharm, tech, and “other” companies compared to academia.
* In terms of environmental impact, the data was limited and as a result the  quantitative analysis of the environmental impact of  machine learning research in biology and chemistry was limited.

**Summary Of The Review:**


The paper presents the critical interfaces of social and environmental impact of machine learning research specifically in chemistry and biology.  The findings reported in the paper are applicable to other fields as well.  This work provided a basis for discussion as key fields such as drug development, genetics, and cancer research can be affected as well. Overall, this overview provided essential knowledge for the advancement of machine learning research in biology and chemistry.

---

### Official Review · Reviewer_dCqg · 2022-10-25

**Confidence:** 3
**Correctness:** 3
**Technical Novelty And Significance:** 1
**Empirical Novelty And Significance:** Not applicable
**Recommendation:** 5

**Clarity, Quality, Novelty And Reproducibility:**

Clarity is good, although the introduction abruptly dives into the
history of hardware development.  The paper uses existing techniques
but the novelty mainly lies in applying the analyses to research in
biology and chemistry.




**Strength And Weaknesses:**

Strengths:

-- A variety of useful insights and takeaways into how  ml (in particular large-scale deep ml) is impacting other
   scientific fields, in particular biology and chemistry, and several
   trends are estimated from analysis of published research  (societal/environmental/energy impact).

Weaknesses:

-- No or very little technical contributions.  The ICLR venue may not to be
   relevant for this report.


**Summary Of The Paper:**


The authors go over trends in the research and use of machine learning
and in particular deep learning in the applied sciences, via analysis
of the open-access literature (bibliometrics and/or full-text analyses),
and highlight areas for concern. The findings are somewhat similar to
prior results of the analyses of recent trends in machine learning R&D
and its related fields.


**Summary Of The Review:**

The paper reports on trends in ml and how it has been developing in
that past decade, specially in biology and chemistry.  Several of
these trends are certainly concerning (the de-democratizatin of ml
work).

However, I don't think that ICLR is the venue for this publication,
just as literature reviews are highly useful, but not relevant in this
conference. The paper does not have a technical contribution.

---

> ### Author Response · Authors · 2022-11-12
> **ICLR as a venue for the publication and introduction**
>
> Dear reviewer,
>
> Thank you for the time spent. Please find some comments and clarifications below.
>
> ### Strength And Weaknesses
> #### Weaknesses
> We decided to submit this paper, which we initially would have agreed to not fit into what we had commonly read at ICLR, because we found the list of *relevant topics explored at the conference* on the [conference page](https://iclr.cc/). This list includes the item **societal considerations of representation learning including fairness, safety, privacy, and interpretability, and explainability**. Based on this, we made the decision to submit the manuscript to ICLR, as I think it makes a relevant contribution to this topic. Let us know if you understand this differently.
>
> ### Clarity, Quality, Novelty And Reproducibility
> We agree on the abrupt dive into hardware. In order to amend this, we added an introductory paragraph to the manuscript that acts as an introduction to the paper on its structure.
>
> We hope that you would reconsider your recommendation based on this feedback. You can find all the changes mentioned in the updated pdf document.
>
> The authors.

---

> > ### Comment · Reviewer_dCqg · 2022-11-18
> > **Acknowledging authors' response**
> >
> > I have read all the reviews and author responses.  My understanding of the call for papers  for that subtopic would be a fairly technical contribution that could address a societal concern such as bias.  However, I can see authors' interpretation.  I have increased my rating as the authors also improved the presentation (adding an intro).  I am borderline, considering overall novelty and contribution.

---

### Decision · Program_Chairs · 2023-01-20

**Decision:**

Reject

**Justification For Why Not Higher Score:**

The reviewers pointed out to very limited technical contributions and minimally coherent article structure. The authors also ignored one reviewer’s comments on the weaknesses. Even though that reviewer gave a high score, the pointed weaknesses still need to be addressed. Overall, there still seems to be a lack of enough enthusiasm among the reviewers after the author responses.

**Justification For Why Not Lower Score:**

N/A

**Metareview: Summary, Strengths And Weaknesses:**

The authors went over trends in the research and use of machine learning and in particular deep learning in chemistry and biology, via analysis of the open-access literature (bibliometrics and/or full-text analyses), and highlight areas for concern. Some of the authors' observations include a potential inequity increase in applied machine learning research in biology and chemistry, and big companies tend to be actively involved in machine learning based research. The findings are somewhat similar to prior results of the analyses of recent trends in machine learning R&D and its related fields. The reviewers pointed out to very limited technical contributions and minimally coherent article structure. The authors also ignored one reviewer’s comments on the weaknesses. Even though that reviewer gave a high score, the pointed weaknesses still need to be addressed. Overall, there still seems to be a lack of enough enthusiasm among the reviewers after the author responses.